# Tailoring Epoxy Network Architecture and Stiffness-Toughness Balance Using Competitive Short- and Long-Chain Curing Agents: A Multiscale Simulation Study

**DOI:** 10.3390/polym17101297

**Published:** 2025-05-09

**Authors:** Zhiyong Dong, Yuqing Li, Renhai Huang, Xuze Zhang, Mingyang Li, Duo Liu, Rui Shi, Xuanbo Zhu, Jianxin Mu, Hujun Qian

**Affiliations:** 1State Key Laboratory of Supramolecular Structure and Materials, Institute of Theoretical Chemistry, College of Chemistry, Jilin University, Changchun 130012, China; dongzy21@mails.jlu.edu.cn (Z.D.); yuqingl23@mails.jlu.edu.cn (Y.L.); huangrh24@mails.jlu.edu.cn (R.H.); zhangxz20@mails.jlu.edu.cn (X.Z.); lmy23@mails.jlu.edu.cn (M.L.); liuduo24@mails.jlu.edu.cn (D.L.); shirui816@jlu.edu.cn (R.S.); 2Key Laboratory of High Performance Plastics, National & Local Joint Engineering Laboratory for Synthesis Technology of High Performance Polymer, College of Chemistry, Jilin University, Ministry of Education, Changchun 130012, China; jianxin_mu@jlu.edu.cn

**Keywords:** epoxy resin, curing agents, network topology, stiffness–toughness balance, multiscale simulation

## Abstract

Designing high-performance crosslinked polymers requires overcoming the inherent stiffness–toughness trade-off through precise control of the network topology. Using epoxy resin as a model system, we establish a multiscale simulation framework to investigate curing reaction kinetics, network evolution, and structure–property relationships. By employing m-phenylenediamine (mPDA) and 1,3-bis(3-aminophenoxy)benzene (DABPB) as competing short- and long-chain curing agents, we demonstrate how network architecture dictates mechanical performance. Simulations reveal that mPDA produces a dense, heterogeneous network with enhanced stiffness, whereas DABPB forms a more uniform structure with greater chain mobility, leading to improved toughness. Through stoichiometric tuning, we achieve fine control over crosslink density and mechanical properties. Furthermore, we decouple cavity formation mechanisms into pendant chain slippage and bond rupture, offering molecular-level insights for the rational design of epoxy resins with programmable mechanical behavior.

## 1. Introduction

The mechanical properties of polymer networks are central to their functionality, enabling their widespread use in diverse engineering and high-performance applications [1,2,3,4]. Designing networks with exceptional and tunable mechanical properties is crucial, both from a fundamental scientific perspective and for practical applications [5]. These properties are governed by the highly crosslinked network structure, which depends critically on monomer composition and polymerization conditions [6,7].

Epoxy resins, a classic class of thermosetting polymers, consist of epoxy-containing base resins and curing agents with active hydrogens (e.g., amines). Upon mixing and heating, these components react to form a three-dimensional crosslinked network. The resulting network structure is dictated by the molecular architectures of the resin and curing agent. Tuning curing agent properties—such as chain length [8], backbone rigidity [9], reactivity, and topology—enables precise control over network mechanics. Designing high-performance epoxy resins requires a deep understanding of network formation mechanisms. However, the complexity of multi-reactive-site networks [10,11,12] and the lack of effective experimental characterization tools pose significant challenges. At the nanoscale, techniques like multiple-quantum nuclear magnetic resonance (MQ NMR) [13,14] and network disassembly spectrometry (NDS) [15,16,17] are limited in resolving fine network structures. Moreover, the insolubility and infusibility of cured epoxies [18] hinder direct structural analysis. Thus, experimental approaches alone are insufficient to unravel network properties. The relationship between synthesis conditions and network microstructure remains poorly understood, leaving crosslinked polymers as a “black box.” In contrast, multiscale molecular dynamics simulations [19,20] offer insights into reaction kinetics and network topology, guiding the design of high-performance materials.

In this work, using diglycidyl ether of bisphenol F (DGEBF) epoxy resin and a 1,3-bis(3-aminophenoxy)benzene (DABPB)—m-phenylenediamine (mPDA) mixed curing system, we develop a chemically specific multiscale simulation strategy to study the curing process. Our approach integrates density functional theory (DFT) calculations [21] with a stochastic coarse-grained (CG) reaction mode [22]. The model combines bottom-up structure-specific iterative Boltzmann inversion (IBI) bonding potentials [23,24] with top-down Lennard-Jones non-bonded interaction potentials [25]. A custom stochastic polymerization model, parameterized by DFT-derived reaction probabilities, captures reactivity differences among DGEBF, DABPB, and mPDA. By varying the DABPB-mPDA ratio, we elucidate how curing agent composition governs network architecture and mechanical properties. Triaxial tensile testing quantifies stiffness and toughness, while structural analyses reveal molecular mechanisms linking network topology to mechanical performance. Our results demonstrate that monomer reactivity and curing agent size dictate crosslink density and chain mobility, enabling tunable properties. Finally, we decouple void formation into pendant chain slippage and bond rupture, providing a theoretical framework for tailoring epoxy networks with programmable properties.

## 2. Materials and Methods

### 2.1. Multiscale Simulation Models for Polycondensation Process

#### 2.1.1. CG Model

To accurately replicate the experimental synthesis process, we explicitly consider the monomers DGEBF, DABPB, and mPDA in our simulation. As shown in Figure 1, the biphenyl structure of DGEBF is mapped to a CG particle labeled as “A”, while the glycidyl ether group is represented by a bead labeled as “B”. The meta-aminophenyl groups in DABPB are mapped to CG particles labeled as “X”, and its meta-diphenyl ether structure is represented by beads labeled as “Y”. For mPDA, it is represented by two “Z” beads, which share the central phenyl ring. It is important to note that mPDA is divided into two beads to account for the reactivity differences between the two hydrogen atoms on the same amine group during the curing process, thus allowing for a more accurate representation of the four amino hydrogens’ reactivity.

The CG intramolecular potentials between different beads are developed by the so-called IBI method. Based on all-atom (AA) simulations, a 4-1 epoxy oligomer (comprising four DGEBF molecules linked via curing agents; see Appendix A) was selected as the crosslinked monomer to extract the target bond length and bond angle probability distributions. The choice of an oligomer rather than a fully crosslinked system was driven by two considerations: first, the structural diversity of crosslinked systems (including crosslinked, dangling, and free molecules) makes the setup of control files for the IBI method exceedingly complex; second, the crosslinking points constrain the conformations of epoxy molecules, placing them in restricted or unrelaxed states and introducing uncertainties in the target probability distributions. In the AA simulations, using the OPLS-AA all-atom force field [26,27], the system contained 100 DABPB and 100 mPDA molecules, forming 200 4-1 epoxy oligomers. The simulations were conducted at 493 K for 200 ns under a pressure of 1 atm, with a time step of 1 fs. Intermolecular interactions were described by a Lennard-Jones (12-6) potential, calibrated to reproduce the density at ambient temperature [28,29]. During the initial equilibration process, a WCA non-bonded potential was applied before monomer reactions to ensure good dispersion. Details regarding both AA and CG simulations can be found in the Appendix A. The results of the CG bonded and non-bonded interaction potentials are supplied in Appendix A.

#### 2.1.2. Simulated Polycondensation Procedure

As outlined in the Introduction, we employed a custom-developed stochastic polymerization model [22] to realistically replicate the curing process of epoxy resins, with reaction probabilities determined by DFT calculations. In the simulations, a reactive shell (*r*_c_ = 0.7 nm) was defined around the growing active chain ends to capture nearby monomer beads. The value of *r*_c_ was determined based on the first peak positions in the non-bonded radial distribution function curves of B-X and B-Z interactions (see Appendix A). Periodic boundary conditions were applied, and an integration time step of 1 fs was used. For the CG simulations of epoxy resin curing, the initial configuration was equilibrated in the NVT ensemble for 15 ns at 493 K, using a Nosé–Hoover thermostat [30,31] with a coupling time of 0.05 ps. The curing reaction then commenced, with a conversion rate of 90% set as the endpoint criterion, consistent with the literature reports [32,33], indicating that the curing process is considered complete at this stage. All simulations were performed using the GALAMOST software v 1.0package [34], with further details on the polymerization process provided in the Appendix A.

## 3. Results and Discussion

### 3.1. Competing Reactions in the Epoxy Resin Curing Process and Rate-Determining Steps

During the curing process, the liquid mixture of epoxy resin and curing agent monomers undergoes chemical reactions to form a crosslinked network. Using amine-based curing agents as an example (Figure 2), the reaction begins with epoxy groups reacting with primary amines, producing crosslinked structures containing hydroxyl and secondary amine groups through addition reactions. Subsequently, the secondary amines further react with epoxy groups from other molecules, forming larger crosslinked structures with hydroxyl and tertiary amine groups.

The addition reaction between amines and epoxy groups is initiated by the nucleophilic attack of amine nitrogen on the terminal carbon of the epoxy ring. Ehlers et al. [35] demonstrated that a stepwise pathway is kinetically more favorable than a concerted mechanism (Figure 1a). These pathways are termed “cyclic” (concerted) and “non-cyclic” (stepwise) based on transition state structures. Hydrogen bond donors, including unreacted amines and reaction-generated alcohols, significantly catalyze the process, with alcohols being the most effective.

This study systematically examines the kinetics of concerted, stepwise, and alcohol-catalyzed pathways using primary and secondary amines. Potential energy surfaces are shown in Figure 1b,c. A bimolecular intermediate (Int_1_) forms via van der Waals interactions, aligning reactants to promote the reaction. The transition state (TS) represents the energy barrier, defined as the Gibbs free energy difference between TS and reactants (Reactant). Int_2_ is an intermediate complex of two product molecules, and Product denotes the final state. Solid lines represent reactions with the long-chain curing agent DABPB, while dashed lines depict the short-chain agent mPDA. The green, blue, and red curves indicate concerted, stepwise, and alcohol-catalyzed mechanisms, respectively. The concerted pathway is kinetically unfavorable due to its high energy barrier (>60 kcal/mol). In contrast, hydroxyl groups in the stepwise pathway enhance reactivity, and hydrogen bonding in the alcohol-catalyzed mechanism drastically reduces the energy barrier, making it the most favorable. Thus, the stepwise/alcohol-catalyzed pathway is the kinetically optimal mechanism.

Table 1 summarizes the Gibbs free energy barriers and reaction rate constants for elementary reactions in the stepwise/alcohol-catalyzed pathway during the curing process, comparing long-chain curing agent DABPB and short-chain curing agent mPDA. For the conversion of primary to secondary amine, DABPB exhibits a free energy barrier of 48.27 kcal/mol and a reaction rate constant of 1.90 × 10^−7^ s^−1^Lmol^−1^, while mPDA shows a lower barrier (46.95 kcal/mol) and a higher rate constant (7.33 × 10^−7^ s^−1^Lmol^−1^), indicating faster reactivity. Similarly, for the conversion of secondary to tertiary amine, DABPB has a barrier of 49.24 kcal/mol and a rate constant of 7.05 × 10^−8^ s^−1^Lmol^−1^, whereas mPDA has a barrier of 46.84 kcal/mol and a rate constant of 8.09 × 10^−7^ s^−1^Lmol^−1^. Overall, the reaction rate of primary-to-secondary amine conversion with mPDA is 3.85 times faster than with DABPB, and this rate advantage increases to 11.48-fold for the secondary-to-tertiary amine conversion. Notably, the primary amine conversion rate of DABPB is 2.70 times faster than its secondary amine conversion, indicating progressively slowing reaction kinetics. In contrast, mPDA exhibits a 10% acceleration in secondary-to-tertiary amine conversion relative to its primary-to-secondary conversion, demonstrating a self-accelerating kinetic profile. This marked contrast in reaction dynamics highlights the superior reactivity of mPDA.

In CG simulations, reaction probabilities are set based on rate constant ratios. For DGEBF epoxy resin with DABPB, the probability of secondary-to-tertiary amine conversion (P_r_(DABPB, S→T)) is 1.00 × 10^−6^, and primary-to-secondary amine conversion (P_r_(DABPB, P→S)) is 2.70 × 10^−6^. For DGEBF with mPDA, these probabilities are 1.04 × 10^−5^ (P_r_(mPDA, P→S)) and 1.15 × 10^−5^ (P_r_(mPDA, S→T)), respectively. Quantum chemical calculation details are provided in the Appendix A.

### 3.2. Simulated Curing of Epoxy Resin

In this study, the system contained 20,000 DGEBF molecules and 10,000 curing agent monomers to maintain a 1/1 stoichiometric ratio between epoxy groups and amine hydrogens. The system size was rigorously selected based on prior computational evidence, demonstrating that thermodynamic and mechanical properties converge when epoxy systems exceed 1250 epoxy monomers and 625 curing agents [36]. Our model, being an order of magnitude larger (20,000/10,000 monomers), ensures the elimination of finite-size effects while providing sufficient spatial heterogeneity for robust cross-linked network development—a critical prerequisite for reliable mechanical property evaluation. To explore the impact of curing agent composition on network formation, five DABPB/mPDA ratios were tested: 1/0, 3/1, 1/1, 1/3, and 0/1. The mixture was equilibrated at 493 K for 15 ns under non-reacting conditions, followed by curing until 90% conversion was achieved. Post-curing, triaxial tensile deformation simulations at 300 K were performed to measure mechanical properties, including Young’s modulus, yield strength, tensile strength, and toughness. The system was stretched to 300% strain at a strain rate of 3 × 10^9^ s^−1^, with constant cross-sectional area. Notably, the network contains only C-C bonds and newly formed C-N bonds, with the latter exhibiting lower bond energy. Consequently, bond rupture during subsequent stretching only occurs for the C-N bonds when their length exceeds 1.5 times the equilibrium bond length. We monitor C-N bond lengths every 10 fs, and delete any C-N bonds surpassing this threshold, along with their associated angles. Young’s modulus was derived from the linear slope of stress–strain data (ϵ ≤ 0.02). While yield strength, tensile strength, and toughness were calculated as average stresses at ~10% strain, maximum stress regions, and the area under the stress–strain curve, respectively. Void volume and number were quantified to characterize deformation-induced voids. Notably, in the CG molecular dynamics simulations conducted in this study, a unit reduction scheme was implemented to bridge the dimensionless Lennard-Jones potential with real physical quantities. The fundamental reduced units were defined as length (σ), energy (ε), and mass (m). From these base units, derived quantities were obtained including time (τ = √(mσ^2^/ε)), pressure (ε/σ^3^), and force (ε/σ). For stress–strain analysis, the simulated stress values were converted to physical units by multiplying with the reduced pressure unit (16.388 atm), while temperature scaling was maintained using the Boltzmann constant (0.008314 kJ·mol^−1^·K^−1^). This systematic reduction approach ensures direct comparability between simulation results and experimental data.

As shown in Figure 2, amine groups in the crosslinking process are categorized as primary (-NH_2_), secondary (-NRH), and tertiary (-NRR) amines. Crosslinking ceases when -NRR reaches 0.8 due to excessive internal energy, preventing further bond formation. Since all five DABPB/mPDA stoichiometric ratios follow the same trend, only the 1/0 and 0/1 cases are presented. As -NRR increases, primary amines decrease, while secondary amine conversion first rises, peaking at 40%-NRH, before declining. Tertiary amines remain inactive until secondary amine conversion peaks, after which they rapidly increase. The distinct inflection point at 40% -NRH marks the point where the three-dimensional network starts to grow.

The defects in the epoxy resin network can be categorized into three main types: (1) residual unreacted monomers, (2) pendant chains, and (3) double bonds generated through a 1 + 1 intramolecular ring-closing reaction occurring between a single epoxy molecule and a curing agent, as illustrated in Figure 3. Detailed defect data are provided in Table 2. At a DABPB/mPDA ratio of 0/1, residual monomers and pendant chains are minimized due to mPDA’s exceptional reactivity, while cyclic defects (double bonds) reach their maximum concentration through intramolecular epoxy-amine cyclization. At DABPB/mPDA = 0/1, the system has a minimum content of residual unreacted monomers and pendant chains, due to mPDA’s high reactivity. In contrast, the fraction of cyclic defects (double bonds) is maximized, as intramolecular cycles form via direct epoxy-curing agent reactions. Note that higher mPDA content will result in a higher mass density of the product (Appendix A) due to its smaller size. We also speculate that higher content of mPDA will result in a pronounced heterogeneity, detailed discussions can be found in the following section.

In epoxy resin crosslinked networks, cyclic structure characterization is crucial for the understanding of material properties. Key parameters include cycle size and cycle efficiency [37]. The former is defined as the number of constituent units (both epoxy and curing agents) forming the closed cycle, while the latter reflects the degree of network connectivity. As illustrated in Figure 3a,b, cycle efficiency is defined as$$\gamma =\frac{n}{L}$$
where γ is the cycle efficiency, L cycle size, and n is the cycle order$$n=n_{c}+\frac{n_{f}}{2}$$
where $n_{c}$ is the number of curing agents where all amine H atoms are all reacted, and $n_{f}$ is for the one for which only three amine H atoms are cured. For the latter, its cycle efficiency is only half of the former.

Figure 3c presents the distribution of cycle efficiency for cycles with L ≤ 10 under varying DABPB/mPDA ratios. As the fraction of the short curing agent mPDA increases, cycle efficiency rises, consistent with its higher reactivity from DFT calculations. This promotes more complete crosslinking and a denser network. Conversely, increasing the proportion of the long curing agent DABPB lowers cycle efficiency, leading to a sparser network. Thus, we predict that higher mPDA content enhances crosslinking density, resulting in a stiffer cured product. Evidence can also be found in the analysis of number of cycles during the curing process, detailed results can be found in Appendix A.

To verify the homogeneous dispersion of each component within the cured network, we calculated the radial distribution functions of molecular centroids for different components at a DABPB/mPDA stoichiometric ratio of 1/1 (Figure 4). The results demonstrate uniform spatial distribution without any aggregation phenomena, further confirming that both long- and short-chain curing agents maintain well-dispersed states throughout the network. Results for other systems are shown in Appendix A.

The stress–strain curves in Figure 5a can be divided into two distinct deformation regimes. During initial deformation (0–100% strain), the pure mPDA system (DABPB/mPDA = 0/1) demonstrates superior Young’s modulus and toughness, which can be attribute to its higher crosslinking density. These mechanical properties decrease progressively with increasing DABPB content. Remarkably, this trend reverses at a strain of roughly 100%, especially for system with pure DABPB exhibits an obvious strain hardening effect after a strain of 100%. This mechanical behavior correlates with the evolution of cycle structures shown in Figure 5b, where shows an obvious breakage of cycle units at a strain of 50%, responsible for a faster mechanical failure. In contrast, for the pure DABPB system (1/0), the breakage of cycle structures is postponed to a larger strain of ~125%. The fracture points denoted by crosses in Figure 5a provide compelling evidence for these performance differences. Detailed mechanical property data are provided in Table 3.

Figure 6a illustrates the representative initial three cyclic structures in systems with pure DABPB (XYX) or mPDA curing agents (Z). To accurately represent size differences, mPDA was modeled as a single Z particle (blue dashed circle). Apparently, the cycle size in DABPB system increases by 2 bead units per step, whereas mPDA systems exhibit 4-bead-unit increments. Figure 6b displays the distributions of cyclic structure sizes in different systems. Pure DABPB system has a rather uniform distribution in cycle size, while systems with higher mPDA content exhibit more heterogeneous distribution, i.e., there are larger distributions for larger cycles. It can be primarily attributed to the higher reactivity of short mPDA, which leads to faster reaction rates and more uneven cyclic structure size distributions. Figure 6c,d document the changes in the number of cycle structures for pure DABPB (1/0) and pure mPDA (0/1) systems between 100% and 200% strains. The results are consistent with that in Figure 5b, i.e., system with pure mPDA exhibits a faster rupture of cycle structures during deformation.

The microporosity of the polymer matrix was characterized using a mesoscale grid analysis method. The simulation domain was discretized into cubic unit cells (2σ × 2σ × 2σ), where voids were identified as cells unoccupied by polymer segments. This methodology, based on geometric occupancy principles, effectively identifies mechanically vulnerable regions following established protocols [38,39]. Figure 7 summarizes the deformation behavior of systems with different DABPB/mPDA ratios: (a) snapshots of configurations and corresponding void distributions at a triaxial strain of 250% (voids shown in blue); (b) evolution of the total void volume; and (c) variation in the number of voids during the deformation process. As shown in Figure 7c, the void count stabilizes or slightly increases at a strain of 100% (first dashed line) and remains relatively unchanged until ~200% strain. During this stage, however, the total void volume continues to rise (Figure 7b), indicating that the increase originates from the expansion of pre-existing voids rather than the formation of new ones. Beyond 200% strain, the void number decreases, which can be attributed to void coalescence and the resulting structural failure. Although the total void volumes at fracture (~200% strain) are similar across different systems, those with higher DABPB content display significantly greater void counts. This more uniform void distribution is favorable for stress dissipation, thereby enhancing the material’s fracture resistance. These findings are consistent with the mechanical performance data, further confirming that long-chain DABPB curing agents play a critical role in improving damage tolerance through more homogeneous void dispersion.

Void formation during tensile deformation proceeds via two primary mechanisms: (1) slippage between topologically non-contributing pendant chains, driven primarily by weak non-bonded interactions (Figure 8a); and (2) bond fracture-induced void generation (Figure 8b). As shown in Figure 8c, increasing the proportion of the long-chain curing agent DABPB leads to a more uniform spatial distribution of pendant chains, thereby promoting more homogeneous void formation during deformation. In contrast, systems with higher mPDA content exhibit a more heterogeneous distribution. Figure 8d reveals that at 200% strain, systems dominated by DABPB show more evenly distributed bond fracture points, facilitating the formation of regular void structures. These findings fully explain the microscopic void formation mechanism observed in Figure 7a.

Notably, Figure 8d also reveals that bond fracture events in all systems begin to occur within a narrow spatial range of <0.3 nm, with the average spacing between fracture points decreasing progressively as the mPDA content increases. To clarify this phenomenon, Figure 9 compares the evolution of amine group types in two stoichiometrically extreme systems—pure DABPB (1/0, Figure 9a) and pure mPDA (0/1, Figure 9b)—during tensile deformation. In the pure DABPB system, the counts of primary, secondary, and tertiary amines remain essentially unchanged within the 0–100% strain range. In contrast, the mPDA system exhibits a noticeable decrease in tertiary amines, along with a concomitant increase in primary and secondary amines, indicating that the bond cleavage events have gradually started. At 200% strain, both systems show a pronounced reduction in tertiary amines accompanied by the formation of primary amines, suggesting a sequential cleavage pathway (–NR_1_R_2_ → –NR_1_– → –N–). This transformation supports the interpretation that the shorter molecular length of mPDA facilitates closer packing and consequently shorter bond fracture spacing at higher concentrations, in contrast to the more extended DABPB chains.

## 4. Conclusions

This study establishes a multiscale simulation framework by integrating density functional theory (DFT) with a coarse-grained (CG) reaction model to investigate the curing process of diglycidyl ether of bisphenol F (DGEBF) epoxy resin systems, with a focus on the influence of mixed curing agents, m-phenylenediamine (mPDA) and 1,3-bis(3-aminophenoxy)benzene (DABPB). The results show that mPDA, owing to its higher reactivity and smaller molecular size, leads to the formation of a highly crosslinked, rigid network that enhances the material’s modulus. In contrast, DABPB, with its longer flexible chains and lower reactivity, promotes the formation of a topologically uniform network that enhances energy dissipation, thereby improving toughness and deformation tolerance.

By varying the stoichiometric ratio of mPDA to DABPB, the network structure can be systematically tuned which, in turn, modulates key mechanical properties, including Young’s modulus, yield strength, tensile strength, and toughness. Structural analysis demonstrates that the curing agent composition plays a crucial role in determining the network architecture. We have decoupled the void formation mechanism during tensile deformation into two processes: the relative slippage of topologically non-contributing pendant chains and bond breakage, successfully elucidating the underlying mechanisms. These findings provide a deeper understanding of the role of curing agents in network formation, offering theoretical guidance for the precise control of epoxy resin structure and properties through the careful selection and combination of curing agents, ultimately enabling performance optimization for diverse engineering applications.

## Data Availability

The original contributions presented in this study are included in the article. Further inquiries can be directed to the corresponding authors.

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
