# Peer review of "Tailoring Epoxy Network Architecture and Stiffness-Toughness Balance Using Competitive Short- and Long-Chain Curing Agents: A Multiscale Simulation Study"

_polymers, 2025, doi:10.3390/polym17101297_

Round 1
Reviewer 1 Report
Comments and Suggestions for Authors
The authors developed a multiscale simulation strategy integrating DFT calculations with a stochastic CG reaction model to study the curing process of diglycidyl ether of bisphenol F (DGEBF) epoxy resin and a 1,3-bis(3-aminophenoxy)benzene (DABPB) - m-phenylenediamine (mPDA) mixed curing system. Their findings revealed that the smaller molecular size and high reactivity of mPDA lead to more crosslinking, thereby enhancing Young's modulus. On the other hand, the longer, flexible chains of DABPB exhibit lower reactivity but form a more uniform network, which improves energy dissipation and results in higher toughness and deformation tolerance. The authors highlight the role of void formation and expansion during deformation on the mechanical properties of cured resin. They explain the void formation as driven by two mechanisms 1) relative slippage between chains interacting through non-bonded interactions and 2) bond breakage. Their results indicate that DABPB systems exhibit uniform void distribution, whereas mPDA systems show a heterogeneous void distribution. Finally, the authors claim that by tuning the DABPB/mPDA composition as curing agents, the epoxy resin properties can be precisely controlled and designed for specific purposes. The authors have employed a robust multi-scale computational framework to investigate the crosslinking behavior and the resulting mechanical properties of epoxy resin cured with both short- and long-chain curing agents. They performed adequate analysis and provided sound explanations to explore the underlying mechanisms and structural variations among the systems considered in this study. I think this manuscript is acceptable for publication in its current form.
Reviewer 2 Report
Comments and Suggestions for Authors
This manuscript presents a multiscale simulation study investigating the curing kinetics, network formation, and mechanical behavior of epoxy resins cured with a mixture of short- and long-chain amine curing agents. Using a combination of DFT calculations and coarse-grained MD, the authors report how network architecture can be tailored to overcome the traditional stiffness–toughness trade-off in crosslinked polymers. The topic is an important materials design problem and offers interesting insights. However, there are critical issues concerning broader contextualization that must be addressed before the manuscript can be considered for publication.
1/ Please discuss how your predicted modulus and toughness values compare with available experimental data for similar DGEBF–amine systems.
2/ Provide error bars or standard deviations for all mechanical property values presented in Table 2.
3/ Elaborate on how the choice of 90% conversion impacts the observed network properties.
4/ Can the authors please clarify the cycle efficiency concept with a more intuitive visual example and explanation.
5/ I would suggest the authors to consider including sensitivity analysis for strain rate and simulation box size effects.
6/ Can the authors discuss how catalyst concentration or hydroxyl content (beyond stoichiometry) might affect the reaction pathways and network evolution.
7/ I think it is very important to quantify the contribution of pendant chain slippage versus bond rupture to the overall void formation.
8/ I suggest the authors to validate the void detection methodology or at least discuss its limitations briefly.
Reviewer 3 Report
Comments and Suggestions for Authors
In the study by Dong et al., the authors investigated the structural and mechanical properties of crosslinking polymer networks using multiscale computer simulations. They used a state-of-the-art approach developed based on DFT and coarse-grained molecular dynamics simulations. The manuscript is well written and contained novel results. The paper could be considered for publication in Polymers after addressing some issues.
- In the simulation methods, additional information about the all-atom force field type should be added.
- In Table 2, the error bars for mechanical properties should be shown.
Round 2
Reviewer 2 Report
Comments and Suggestions for Authors
The revised version looks good.